# Development of the Readiness for Home-Based Palliative Care Scale (RHBPCS) for Primary Family Caregivers

**DOI:** 10.3390/healthcare9050608

**Published:** 2021-05-19

**Authors:** Meng-Ping Wu, Lee-Ing Tsao, Sheng-Jean Huang, Chieh-Yu Liu

**Affiliations:** 1Department of Nursing, Taipei City Hospital, Taipei 103, Taiwan; B1800@tpech.gov.tw; 2School of Nursing, National Taipei University of Nursing and Health Sciences, Taipei 112, Taiwan; 3School of Nursing, National Taipei University of Nursing and Health Sciences, Taipei 112, Taiwan (retired at 2019); leeing.tsao@gmail.com; 4Superintendent Office, Taipei City Hospital, Taipei 103, Taiwan; daw15@tpech.gov.tw; 5Department of Surgery, College of Medicine, National Taiwan University, Taipei 100, Taiwan; 6Biostatistical Consultant Laboratory, Department of Speech Language Pathology and Audiology, National Taipei University of Nursing and Health Sciences, Taipei 112, Taiwan; 7Department of Teaching and Research, Taipei City Hospital, Taipei 103, Taiwan

**Keywords:** palliative care, end of life, caregiver, reliability, validity

## Abstract

In Chinese or Eastern society, most end-of-life (EOL) patients still choose to die at home. However, primary family caregivers usually do not prepare themselves to face the death of patients. Therefore, a measurement of the readiness for home-based palliative care for primary family caregivers is needed. In this study, the readiness for home-based palliative care scale (RHBPCS) for primary family caregivers was developed to assess the readiness of primary family caregivers. This study recruited 103 participants from five branches of one municipal hospital system. The reliability and validity of the RHBPCS was evaluated using expert validity examination, confirmatory factor analysis (CFA), and item analysis. The results showed that the RHBPCS had strong goodness-of-fit and good reliability and validity. In summary, the RHBPCS is suggested for assessing the readiness for home-based palliative care of primary family caregivers.

## 1. Introduction

In recent years, there is an increasing trend of choosing home-based palliative care (HBPC) in end-of-life (EOL) patients [1]. To meet EOL patients’ holistic care needs, physical, psychological, and spiritual care-related needs should be taken into account by clinicians. Published studies and reports have shown that HBPC provides care, not cure, in resolving uncomfortable symptoms and relieving pain to ensure that patients can feel more comfortable and have a good EOL at home [2,3,4,5,6,7,8]. Primary family caregivers who provide HBPC are suggested to have professional nursing skills, including symptom observation; emergency handling; and the ability to provide medication and pain relief, including attention to the use of oral morphine to prevent the patient from becoming addicted; however, family caregivers often lack professional training and understanding [9]. Studies have shown that more than half of family caregivers suffer from depression [10]; however, when patients have confidence in their HBPC and the family caregivers equip themselves well, the physical and mental burdens and physical injury to patients can be reduced [11,12], and the health-related quality of life of both patients and family caregivers can be improved.

Family caregivers play an important role for HBPC, and need to have appropriate knowledge; a caring, positive attitude; and maintain their emotional, mental, social, and physical health. However, clinicians can teach family caregivers the value and benefits of HBPC, and can provide necessary help and support for EOL patients at home with their families [13]. As an increasingly prevalent care mode, HBPC has gained increasing importance worldwide, and the readiness of family caregivers is now regarded as an important factor for the promotion of HBPC [14]. However, according to the current knowledge, there is no measurement instruments or scales for measuring the readiness for home-based palliative care of family caregivers. Therefore, this study aimed to develop a novel scale, the readiness for home-based palliative care scale (RHBPCS) for primary family caregivers.

## 2. Materials and Methods

### 2.1. Participants and Setting

This study adopted an exploratory cross-sectional study design. A convenience sampling method was used. From 1 July to 31 December 2018, the primary family caregivers from five branch regional teaching hospitals in the Taipei City Hospital system were referred by social workers or psychologists. Inclusion criteria were as follows: aged ≥20 years, who were the primary family caregiver for an EOL patient, and who were considering receiving HBPC. Exclusion criteria were as follows: caregivers who had severe diseases, including cancer, stroke, end stage renal disease (ESRD), or severe congenital diseases; and caregivers who had originally decided to accept HBPC and then later changed their decisions. The research team of this study contacted the eligible family caregivers and explained the purpose of the study. After the informed consent forms were obtained, the face-to-face interview was conducted at the eligible family caregiver’s home. Each eligible participant needed about 30 min to finish the RHBPCS survey. The sample size estimation was based on Anderson and Gerbing [15] and Ding et al. [16], who proposed sample size ranging from 100 to 150 as the minimum sample size requirement when constructing structural equation models or conducting confirmatory factor analysis.

### 2.2. Instrument Development

The readiness for home-based palliative care scale (RHBPCS) was initialized following the conceptual framework based on the pilot qualitative study from family caregivers of EOL patients and clinicians [14]. Three stages of the development of RHBPCS are described as follows:

Stage 1—qualitative study: the purpose of the first stage of qualitative research was mainly to explore the life experience, caring burden, and concerns of the caregivers during the period of caring for EOL patients. Twenty-two family caregivers (≥20 years old) were recruited (5 males and 17 females), and their experience of caring for end-of-life patients ranged from 2 months to 2 years and 8 months. A semi-structured interview guide was developed, which was based on the grounded theory, and the qualitative data were collected using face-to-face interviews. “Wholeheartedly accompanying one’s family to the end of life at home” was the core category. Six important themes representing caregivers’ experiences were abstracted from the qualitative face-to-face interviews: (1) learning the basic skills of end-of-life home care; (2) arranging the sharing and rotation of care; (3) preparing for upcoming deaths and funerals; (4) negotiating the cultural and ethical issues of end-of-life home care; (5) ensuring a comfortable life with basic life support; and (6) maintaining care characterized by concern, perseverance, and patience [14].

Stage 2—expert validity analysis and constructing RHBPCS: based on the important themes abstracted from the qualitative face-to-face interviews, the first version of the scale assessing the readiness of home-based palliative care was drafted, which was comprised of 6 subscales with 23 items. Two rounds of expert panels were conducted, and each round of the expert panel invited 8 experts: one professor whose expertise was palliative care policy, four physicians whose expertise was end-of-life care, two senior nurses with more than 10 years of working experience in palliative care units, and one caregiver who was caring for an end-of-life patient at home. After 2 rounds of expert panels, the original 7 subscales were integrated into 4 subscales and 8 items were deleted due to either being duplicates of other items or having unclear item descriptions. The revised version of RHBPCS was determined as follows: (1) family maintenance and consensus (items 1–3); (2) home care skills and hospice preparation (items 4–8); (3) arrangements for sharing and rotation (items 9–11); and (4) timely emergency management and palliative care (items 12–15).

Stage 3—quantitative study: 8 experts were invited to check the items for description clarity and item feasibility, and the content validity index (CVI) was 0.953, which indicated a very good content validity [15,16]. Furthermore, a pilot study was conducted, 30 caregivers were recruited to conduct item analysis, and the critical ratios (CR) of the 15 items all showed statistically significant differences, which indicates that each item had good discrimination validity [17]. Confirmatory factor analysis (CFA) was used to examine the goodness-of-fit of the factor structure. Concurrent validity was assessed by examining the correlation of RHBPCS with criterion-related validity analysis using the Chinese version of the carer support needs assessment tool (CSNAT). Cronbach’s α was used to assess the internal consistency reliability.

This study was conducted according to the guidelines of the Declaration of Helsinki and approved by the Taipei City Hospital Research Ethics Committee (protocol code THCIRB-1070410). The end-of-life patients in this study, before entering the end-of-life stage, had been constantly seeking medical treatments in Taipei City Hospital system for a long time when the end-of-life patients were referred by social workers or psychologists to receive home-based palliative care; the researchers of this study had known the family caregivers for a period of time and they had established good relationships with each other. Therefore, the researchers of this study did not provide any financial incentives, all of the participants were willing to participate in this study, and their personal information will not be shared to other researchers or have any patent applied to it.

### 2.3. Data Analysis

The continuous variables were displayed using mean and standard deviation (SD) and the categorical variables were displayed using case number (n) and percentage (%). In addition, the Chinese version of the carer support needs assessment tool (CSNAT) was used as a criterion-related validity index. Confirmatory factor analysis (CFA) was used to test the goodness-of-fit of the conceptual framework and the reliability and validity of the RHBPCS. Before conducting CFA, a pilot study was conducted which recruited 30 participants, and necessary adjustments to the draft of RHBPCS were made. Pearson’s correlation coefficient was used to test the criterion-related validity, and Cronbach’s α was used to test the internal reliability. Statistical analysis was conducted using IBM SPSS version 21, and the CFA was conducted using LISREL version 8.8. A two-tailed significance level of 0.05 was considered statistically significant.

## 3. Results

At the end of the study period (31 December 2018), 130 family caregivers were contacted; however, 27 were excluded due to the death of the end-of-life patients, therefore 103 eligible participants were recruited; the response rate was about 80%. Among the study sample, 33 (32%) were males, 70 (68%) were females, and the mean age was 61.17 years (standard deviation (SD) = 11.62 years). The demographic information of the participants is tabulated in Table 1. The overall content validity index (CVI) of the RHBPCS was 0.953. The finalized version of the RHBPCS is provided in Appendix A.

The RHBPCS is comprised of 15 items in 4 subscales as follows: (1) family maintenance and consensus (items 1–3); (2) home care skills and hospice preparation (items 4–8); (3) arrangements for sharing and rotation (items 9–11); and (4) timely emergency management and palliative care (items 12–15). After the second round of expert examination, the overall mean CVI of the scale (S-CVI/Ave) was 0.953, and, except for items 3, 9, 10, 14, and 15, the mean CVI (I-CVI) was 1.00. The RHBPCS adopted the four-point Likert scale.

The criterion-related validity analysis was conducted using the Pearson correlation coefficient, CSNAT, and RHPCS and r = 0.919 (*p*-value < 0.001), which indicated that the RHPCS score was significantly correlated with the CSNAT score.

The Cronbach’s α of the total RHBPCS was 0.928; for the subscales, the Cronbach’s α was 0.859 for “family maintenance and consensus”, 0.879 for “home care skills and hospice preparation”, 0.875 for “arrangements for sharing and rotation”, and 0.860 for “timely emergency management and palliative care” (see Table 2). The KMO index of the RHBPCS was 0.790, which indicates middling to meritorious adequacy of the sampling [18]. In addition, the results of Bartlett’s test of sphericity in this study showed statistically significant differences from the identity matrix (*p*-value < 0.001), which implied that the factor structure of this study can be accepted.

The path coefficients of all the latent variables in relation to the manifest variables all showed statistical significance (*p*-value < 0.05), and there were no negative values in the error variance. The CFA factor loading ranged from 0.52 to 0.80, which was within the acceptable-to-well range of 0.50–0.95 [19]. The goodness-of-fit indices of the measurement model of the RHBPCS included the following: the root mean square residuals (RMR) = 0.0017, the goodness-of-fit index (GFI) = 0.95, the adjusted goodness-of-fit index (AGFI) = 0.93, the parsimony goodness-of-fit index (PGFI) = 0.67, the parsimony normed fit index (PNFI) = 0.80, the root mean squared error of approximation (RMSEA) = 0.051, the average variance extracted (AVE) = 0.66, and the construct reliability (CR) = 0.96, which showed good goodness-of-fit, implying that the RHBPCS had good validity [19,20]. The path diagram is shown in Figure 1.

The results of item analysis showed that each time showed statistically significant differences between the high RHBPCS score group (upper 27% of participants) and the low RHBPCS score group (lower 27% of participants) (*t*-value of all the items were between 2.540 to 5.284, which indicated that all 15 items showed statistically significant differences), and so all of the items of the RHPCS showed good discrimination validity [21] (see Table 3).

Regarding the item-deleted Cronbach’s α analysis, the results showed that the item-deleted Cronbach’s α ranged from 0.901 to 0.909, compared with the overall Cronbach’s α = 0.928, which implied that if any item was deleted, the Cronbach’s α would decrease; therefore, no redundant item was found in the RHBPCS. Furthermore, the item–total correlation coefficients for each item ranged from 0.522 to 0.753, and all correlation coefficients showed statistical significance, which implied that each item had good item reliability [21] (see Table 4).

## 4. Discussion

In this study, a new self-reported measurement scale, the readiness for home-based palliative care scale (RHBPCS) for primary family caregivers was developed to assess if the primary family caregivers had prepared themselves enough to undertake the home-based palliative care of end-of-life patients. The results of confirmatory factor analysis (CFA), item analysis, and reliability analysis showed that the RHBPCS had good reliability and validity.

Compared with published studies, Allende-Perez et al. [22] showed that the lack of family support and consideration may increase the pressure on caregivers. However, the provision of family support and care can reassure family caregivers and help them to continue to care for end-of-life patients [23]. In addition, a positive attitude towards life and care experience can help caregivers to better cope with the care stress, recognize their deficiencies, seek assistance when patients are in the early end-of-life stage, and prevent increasing the burden of care [24]. The usual care provided by hospitals often does not include family care in the care plan. When the patient’s condition deteriorates or new discomfort symptoms appear, the caregiver may be anxious because of their inability to respond the emergency. If the family caregiver can equip themselves with the skills and confidence to help the end-of-life patients to resolve the symptoms, then they may be able to help reduce the suffering caused by the symptoms, improving the patient’s physical function and helping the patients to maintain a positive attitude towards the care [25,26]. By evaluating, understanding, and enhancing the care skills of caregivers, negative emotions can be reduced, and quality of life (QoL) can be improved [27]. Physical care skills can make patients feel comfortable and keep the body clean and free from odor, which is related to the patient’s self-esteem and body image. At the EOL, the normal appearance of the patient is important [27]. Other aspects, such as the death certificate for the patient, hospice care, the opinions of family members regarding the funeral, and seeking assistance from funeral service companies, were similar to those reported in the study of Glass et al. [28]. In addition to care knowledge and skills, healthcare staff should also consider whether competing needs, such as work, travel, and child care needs, could affect the caregiver’s ability to provide continuous care or overburden them, and appropriate care plans should be discussed with caregivers [29]. To facilitate care, caregivers often live with patients, which can place a heavy physical and mental burden on the caregivers. It is much more reassuring to have family members take turns taking care of patients than to seek assistance from foreign domestic helpers (FDHs), and such rotation of care can give each family member enough breathing room to take care of their own health. In addition, caregivers need time to themselves to allow them to maintain the role as care providers [30]. The psychosomatic symptoms of extended care are associated with the overload caused by the patient’s mental disorder and restlessness [31,32], and the filial duties associated with the long-term critical illness of parents presents a heavy burden. EOL care is difficult to sustain, and appropriate rest and sharing of the care load are important keys to HBPC.

In addition, according to the most up-to-date Cochrane systematic review [33], which indicated that receiving home-based end-of-life care programs may increase the number of people who will die at home, and research that assesses the impact of home-based end-of-life care on caregivers would be useful. The newly developed assessment instrument, the readiness for home-based palliative care scale (RHBPCS), can meet the need for the assessment of the readiness of home-based palliative care for family caregivers. New exploratory qualitative research also indicated that subjective measurement was the most important theme abstracted from 63 healthcare professionals (across 11 specialist palliative care services) [34]. The RHBPCS can be adopted for family caregivers to assess if family caregivers are prepared or ready to take care of an end-of-life patient in the home. Regarding the summary of the RHBPCS, according to the CFA results, the RHBPCS offers sufficient distinction between its various dimensions, and has clinical application value. In addition, the RHPCS was significantly related to the CSNAT, and the items of both scales were correlated, including the need to provide caregivers with physical, technical, social, economic, psychological, and spiritual supports when patients are at the EOL [35,36]. Seow et al. [37] proposed that a good-quality HBPC plan needs to have six important components, four of which are pain and symptom management, the comprehensive management of physical and non-physical symptoms, timely support in emergency situations, and the preparation of the patient and family members, which were included in the RHBPCS scale. Therefore, the RHBPCS not only has good reliability and validity, but also agrees with the mainstream knowledge regarding home-based palliative care.

For future applications, the RHBPCS can be applied by family caregivers to assess if they have prepared themselves well to undertake home-based palliative care, if not, social workers, psychologists, nurses, or physicians are suggested to provide more precise support based on the subscale scores. The RHBPCS also can serve as an important predictor for predicting the status of family caregivers’ mental health in the long term, especially after the death of the end-of-life patient. In the future, because the attitudes facing end-of-life patients and the attitudes of undertaking home-based palliative care may vary from culture to culture , the researchers of this study will continue to develop more educational resources for the RHBPCS for different cultural scenarios, such as majority Christian or Catholic countries, as opposed to Buddhist countries, which can be regarded as one study limitation. The sample size was 103 in this study, which was also regarded as a study limitation in this study because, as the concept of home-based palliative care was proposed in recent years in Taiwan, most family caregivers are still worried that they cannot care for the end-of-life patients well at home. However, Anderson and Gerbing [15] and Ding et al. [16] proposed that a sample size ranging from 100 to 150 is the minimum sample size requirement when constructing structural equation models or conducting confirmatory factor analysis. Therefore, the sample size (*n* = 103) in this study is still acceptable.

## 5. Conclusions

In conclusion, the newly developed RHBPCS had good reliability and validity, and can be recommended for patients, family caregivers, and clinicians to assess if family caregivers are prepared to undertake home-based palliative care. The RHBPCS can help to ensure that patients receive comprehensive care and that family caregivers obtain the ability to care.

## Figures and Tables

**Figure 1 healthcare-09-00608-f001:**
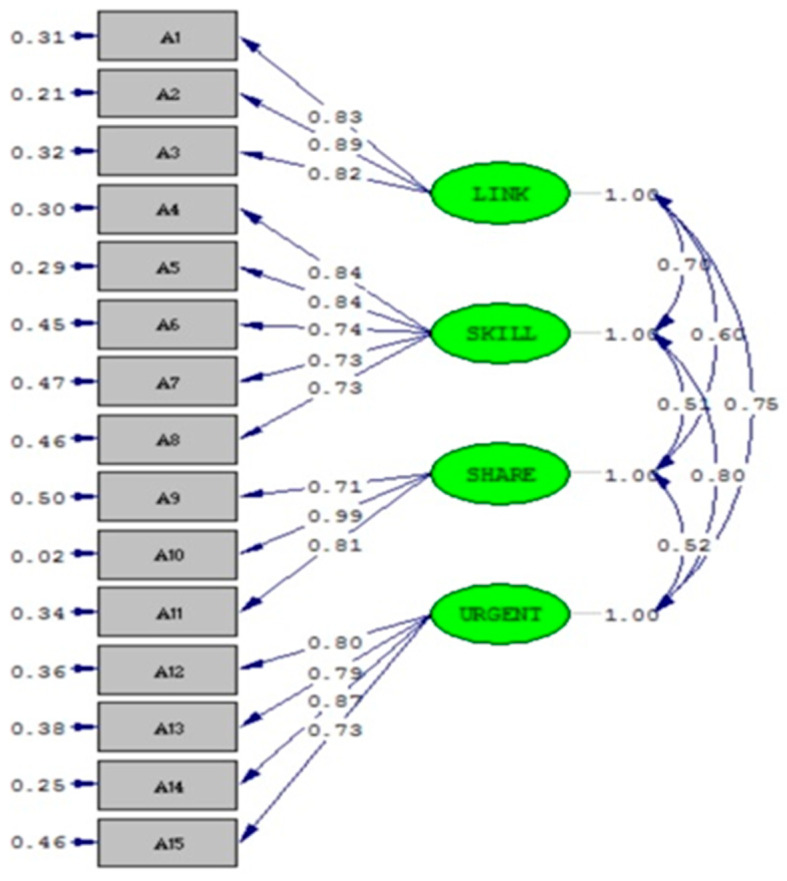
Confirmatory factor analysis results of the RHBPCS.

**Table 1 healthcare-09-00608-t001:** Demographic information of the participants (*n* = 103).

Variable	*n*	(%)
Sex		
Male	33	(32.0)
Female	70	(68.0)
Age		
Under 50 years	18	(17.5)
51–60 years	31	(30.1)
61–70 years	39	(7.9)
71–80 years	9	(8.7)
Above 81 yrs	6	(5.8)
Education		
Under elementary school	14	(13.6)
Junior high school	7	(6.8)
High school	25	(24.3)
Some college or associate degree	57	(55.3)
Married status		
Unmarried	20	(19.4)
Married	75	(72.8)
Divorced	2	(1.9)
Widowed	6	(5.8)
Religion		
Taoism	31	(30.1)
Buddhism	39	(37.9)
Christianity	9	(8.7)
Others	7	(6.8)
None	17	(16.5)
Occupation		
Retiree	37	(35.9)
Homemaker	26	(25.2)
Public sector	6	(5.8)
Business	13	(12.6)
Service industry	17	(16.5)
Others	1	(1.0)
None	3	(2.9)
Caregiver relationship		
Couple	23	(22.3)
Son or daughter	58	(56.3)
Daughter-in-law	13	(12.6)
Other relatives	9	(8.8)
Care time		
<1 year	23	(22.3)
1–3 year	34	(33.0)
≥3 year	46	(44.7)
Couple	24	(23.3)
Son or daughter	14	(13.6)
Other relatives	38	(36.9)
None	27	(26.2)
Chronic disease		
Yes	45	(43.7)
No	58	(56.3)
Patient diagnosis		
Cancer	36	(35.0)
None-cancer	67	(65.0)

**Table 2 healthcare-09-00608-t002:** Internal consistency reliability of the RHBPCS (Cronbach’s α) (*n* = 103).

Sub-Scale	Number of Items	Cronbach’s α
Family maintenance and consensus	3	0.859
Home care skills and hospice preparation	5	0.879
Arrangements for sharing and rotation	3	0.875
Timely emergency management and palliative care	4	0.860
Total scale of the RHBPCS	15	0.928

**Table 3 healthcare-09-00608-t003:** Comparisons of the extreme groups in the RHPCS (*n* = 103).

Items	*t*-Value	*p*-Value
1. Me and patient’s families have reached an agreement and have a thorough understanding of patient’s condition and the home-based palliative care.	**3.959**	**<0.001**
2. I’m fully prepared to arrange the way of rotation.	**3.034**	**0.004**
3.I have time to get enough rest.	**5.014**	**0.000**
4. I know how to manage the patient’s symptoms.	**3.813**	**<0.001**
5. I know how to prepare medications for patient.	**2.540**	**0.014**
6. I know how to make a contact in case of emergency.	**2.553**	**0.014**
7. I can use home medical supplies and equipment for the patient.	**2.780**	**0.007**
8. I have nursing skills to take care the patient.	**3.545**	**0.001**
9. I will follow patient’s wishes, prepare their basal diet and help them with simple exercise.	**2.553**	**0.014**
10. I know the traditional culture and all the preparations for home death.	**4.076**	**<0.001**
11. I am preparing and I know how to manage the signs of death.	**4.666**	**<0.001**
12. I can discussed with the patient about his/her state of an illness and last words.	**5.284**	**<0.001**
13. I am fully prepared for everything in order to take care the patient.	**3.576**	**0.001**
14.I can maintain a caring, perseverance, patient attitude or have a sense of morality and be spiritual supported.	**3.545**	**0.001**
15.I know how to use the caring resources that provided for the home caregiver.	**3.667**	**0.001**

**Table 4 healthcare-09-00608-t004:** Test of internal reliability for the RHBPCS (*n* = 103).

Items	Item Deleted Cronbach’s α	Item-Total Correlation
1. Me and patient’s families have reached an agreement and have a thorough understanding of patient’s condition and the home-based palliative care.	**0.902**	**0.696 *****
2. I’m fully prepared to arrange the way of rotation.	**0.909**	**0.522 *****
3. I have time to get enough rest.	**0.904**	**0.660 *****
4. I know how to manage the patient’s symptoms.	**0.902**	**0.688 *****
5. I know how to prepare medications for patient.	**0.900**	**0.744 *****
6. I know how to make a contact in case of emergency.	**0.904**	**0.660 *****
7. I can use home medical supplies and equipment for the patient.	**0.901**	**0.740 *****
8. I have nursing skills to take care the patient.	**0.903**	**0.671 ****
9. I will follow patient’s wishes, prepare their basal diet and help them with simple exercise.	**0.901**	**0.752 *****
10. I know the traditional culture and all the preparations for home death.	**0.904**	**0.680 *****
11. I am preparing and I know how to manage the signs of death.	**0.900**	**0.753 *****
12. I can discussed with the patient about his/her state of an illness and last words.	**0.905**	**0.678 *****
13. I am fully prepared for everything in order to take care the patient.	**0.907**	**0.551 *****
14.I can maintain a caring, perseverance, patient attitude or have a sense of morality and be spiritual supported.	**0.902**	**0.704 *****
15.I know how to use the caring resources that provided for the home caregiver.	**0.905**	**0.605 *****

**: *p*-value < 0.01; ***: *p*-value < 0.001.

## Data Availability

Not applicable.

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
