# Peer review of "Development of the Readiness for Home-Based Palliative Care Scale (RHBPCS) for Primary Family Caregivers"

_healthcare, 2021, doi:10.3390/healthcare9050608_

Round 1

Reviewer 1 Report

The manuscript presents the results of a study developed a novel scale, Readiness for Home-Based Palliative Care Scale (RHBPCS) for primary family caregivers. The manuscript is well structured. 

The topic is relevant since the valid tools can be extremely useful in improving quality of care of patients and family caregivers.

However, in my opinion, the study presents some methodological problems that need to be addressed and require reorientation on behalf of the researchers. I detail these problems below.

Materials and Methods

Line 71–72 reports that “The Readiness for Home-Based Palliative Care Scale (RHBPCS) was initialized the conceptual framework based on the pilot qualitative study from family care givers of EOL patients and clinicians, also based on a literature review,” however I think that you cannot write literature review if you indicate only one reference. You must include more studies used in the literature review.

Line 73.  You should write the number of experts in every round and their experiences.

Line 100–111: Please, you should indicate the scale.

Results

You should include sociodemographic variables of participants.

Note important:

The references are wrong because in text are 14 (line 53) and the next reference are 23-24. What happen it?

Author Response

Please find the author's reply to the reviewer's comments in the attachment, thank you!

Reviewer 2 Report

The sample size appears to be fairly small for the number(15) of items (generally 5 times of number of items, it means more than 125. Therefore, this limitation have to be mentioned clearly in the discussion section

Please, mention Root Mean squared error of approximation(RMSEA), Average variance extracted (AVE) and Construct reliablilty (CR) also.

Author Response

Please fine the author's reply to the reviewer's comments in the attached file, thank you!

Reviewer 3 Report

The material is interesting and the topic is relevant. The method seems to have been followed faithfully and the authors were well-positioned to conduct the analysis. Despite these positives in my view the paper needs more work before it could be published and I have made some specific suggestions below.

- The abstract in question would benefit from some form of framing of the context of the study, by which I mean the history of the problem and the results already formulated that are relevant to it. This would allow a better understanding of the importance of the topic.

- My major concern pertains to the fact that in its present form the manuscript does not reveal the theoretical assumptions on which scale construction has been based. What theoretical aspects of or basic assumptions on construct had guided the author(s) when they formulated the test items? This should be made explicit, as otherwise the reader might have the impression that items were formulated intuitively.

- Overall organization and clarity throughout the manuscript should be improved. For the introduction, a restructuring of the writing to provide more coherent and connected ideas and sections would be valuable. Brief synopsis or syntheses of ideas and relationship between or within constructs would improve flow dramatically. It is important to specify: how this/your study is increasing our understanding/knowledge regarding the readiness of primary family caregivers to provide home-based palliative care? This would allow a better understanding of the importance of the topic.

METHOD

- There are no sources to help the reader understand the methodological approach taken in the paper (i.e. a mixed [qualitative and quantitative] sequential exploratory design). Because this study was conducted in two phases: the qualitative phase (designing of the questionnaire) and the quantitative phase (assessment of psychometric properties). This needs to be expanded, clarified, and supported by in-text citations.

- The pilot qualitative study needs more detail (item generation based on literature review and interviews).

- More precision is necessary regarding the sampling strategy and access to the target population. Response rate? How were participants recruited? 

- There is no mention of analyses to support sample size adequacy, i.e. Bartlett´s test of sphericity? KMO?

- The time needed to complete the questionnaires is not mentioned, this information is relevant.

- The ethical aspects in collecting data are not specifically clarified, independently of the voluntary nature of the subjects´ participation and the approval by the local IRB; variables such as the offer of incentives to participate (they were compensated for participation), sharing and use of data are not patent. 

RESULTS

- A better visual structure of tables (boldface variables with statistical significance) would improve the readability. 

DISCUSSION

- The discussion appears to relay the main findings, and there is some discussion of what the findings mean, but there was very little attention paid to how this study might relate to our existing knowledge base, and little attention paid to how the current findings might extend our current knowledge.

- In the case of the implications should have been approached in greater depth. Identify recommendations for practice/research/education as appropriate, and consistent with limitations.

EDITORIAL COMMENTS

- The manuscript will serve a broad audience of students, researchers, and practitioners, however, the manuscript needs to be carefully and attentively proofread, because some sentences are awkwardly constructed, punctuation is deficient, and therefore reading is occasionally difficult to follow. The English of this manuscript should be reviewed by a native-English speaker.

- Some of the cited references are somewhat dated.

- Please provide the instrument in supplementary.

Author Response

Please find the author's reply to reviewer's comments in the attached file, thank you!

Round 2

Reviewer 3 Report

The manuscript will serve a broad audience of students, researchers, and practitioners. I appreciate the authors' effort to revise the manuscript. The clarity of the manuscript is much improved.